# Effective Text-to-Image alignment with Quality Aware Pair Ranking

## Abstract

Fine-tuning techniques such as Direct Preference Optimization (DPO) allow one to better align Large Language Models (LLMs) with human preferences. Recent adoption of DPO to diffusion modeling and its derivative works have proven to work effectively in improving visual appeal and prompt-image alignment. However, these works fine-tune on preference datasets labeled by human annotators, which are inherently subjective and prone to noisy labels. We hypothesize that fine-tuning on these subjective preferences does not lead to optimal model alignment. To address this, we develop a quality metric to rank image preference pairs and achieve more effective Diffusion-DPO fine-tuning. We fine-tune using incremental subsets of this ranked dataset and show that diffusion models fine-tuned using only the top 5.33% of the data perform better both quantitatively and qualitatively than the models fine-tuned on the full dataset. Furthermore, we leverage this quality metric and our diverse prompt selection strategy to synthesize a new paired preference dataset. We show that fine-tuning on this new dataset achieves better results than the models trained using human labeled datasets. The code is available at this link.

## 1 Introduction

Currently, diffusion-based Text-to-Image (T2I) (Rombach et al., 2021; Podell et al., 2023; Chen et al., 2023; 2024) models are state-of-the-art in image generation. These models are trained in a single stage on a large-scale dataset of images scraped from the Internet, enabling them to have huge knowledge. However, their outputs often do not align with human preferences as they are not explicitly optimized for this purpose. In contrast, Large Language Models (LLMs) undergo training in two distinct stages: the first stage involves pre-training on large web-scale datasets, while the second stage uses Supervised Fine-tuning (SFT) and Reinforcement Learning based on Human Feedback (RLHF) to align outputs with human preferences. While significant progress has been made in alignment fine-tuning for LLMs, aligning T2I outputs with human preferences remains a difficult challenge.

Recent works have begun exploring how to better align T2I models with human preferences. These approaches can be broadly classified into two broad categories: they either use a reward model trained on human preference data to guide the T2I model or they directly fine-tune the T2I model on pairwise preference data. Reinforcement Learning (RL) based approaches like Alignprop(Prabhudesai et al., 2023), ImageReward(Xu et al., 2023), DDPO(Black et al., 2023) do not scale well to large datasets and are highly prone to problems like overfitting and mode collapse. Additionally, training good reward models and using them to fine-tune diffusion models introduces significant operational challenges as it adds a lot of computational overhead.

To address this gap in diffusion model alignment, approaches like Diffusion-DPO(Wallace et al., 2023a) have emerged, reformulating the loss function to completely remove the reward model and directly fine-tune on paired image preference data, which solves the problems of traditional RL-based approaches. In recent works, more preference alignment approaches such as Diffusion-KTO(Li et al., 2024) and IPO(Azar et al., 2023) have emerged, building on Diffusion-DPO to further improve diffusion model alignment. However, all of these approaches share a common drawback: they either require paired preference data or a label

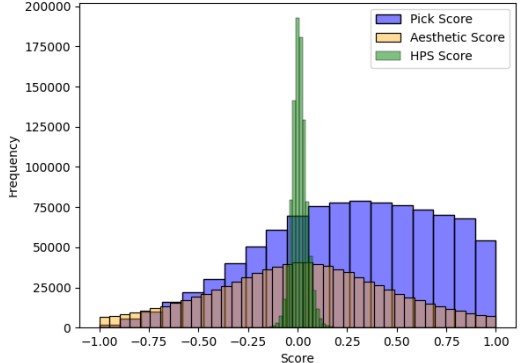
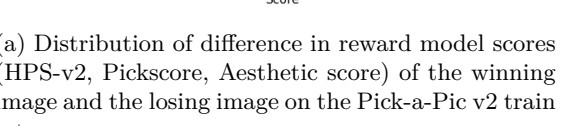

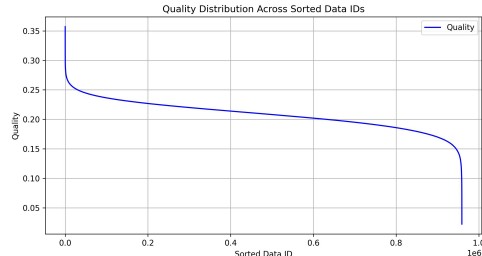

(a) Distribution of difference in reward model scores (HPS-v2, Pickscore, Aesthetic score) of the winning image and the losing image on the Pick-a-Pic v2 train set.

(b) Quality metric plotted in a sorted order for preference pairs in the Pick-a-Pic v2 train set

Figure 1: Left - plot of difference in reward models sores for Pick-a-Pic v2 train set, Right - plot of quality metric on Y-axis with the sorted dataset index on X-axis.

of 'winning' or 'losing' for each image. These labels, collected from human annotators, can be noisy as preference is subjective. Furthermore, these labels do not capture the "strength/quality" of the preference pair and treat each paired sample as equally important, In Figure 1a we plot the distribution of the difference in the reward model scores between the winning and losing images for a paired preference dataset( Pick-a-Pic v2 train set) with three reward models (Aesthetic score, Pickscore (Kirstain et al., 2023), and HPS-v2 (Wu et al., 2023)). As can be observed, the graphs follow an approximately normal distribution with mean around zero, with a large portion of the samples exhibiting higher scores for the "losing" images compared to their assigned "winning" counterparts. Notably, the distribution of PickScore is left-tailed because it has been trained on this subset, however, there are still a big number of samples where PickScore disagrees with the true labels. This suggests a misalignment between the labels given by the human annotators and the scores from the reward models. We hypothesize that samples where the preference scores of both images are close, or even reversed, negatively impact learning. In contrast, focusing on preference pairs where the reward model agrees on a clear winner could lead to more stable and effective fine-tuning. For instance, pairs that focus on individual qualities such as subjective preference, prompt adherence, or image aesthetics might steer the model in different directions, making the learning suboptimal, while fine-tuning on pairs that are consistent across all qualities would result in a better model.

To address these shortcomings, we propose our novel approach — **Effective Text-to-Image Alignment with Quality Aware Pair Ranking**. Specifically, we introduce a quality metric to assess the quality of a preference pair as a fine-tuning sample. We use a carefully devised metric based on the AI reward model score to rank all samples from the alignment fine-tuning dataset. We use ranking to prioritize higher-quality paired samples by fine-tuning on our Quality Sorted Dataset (QSD). We find that training on just the top 5.33% of the QSD yields superior alignment performance compared to training on the full dataset. As the size of the dataset increases, the performance plateaus and then starts to decrease, suggesting that these pairs send ambiguous training signals. We demonstrate through human and AI evaluations that our ranking method improves the performance of state-of-the-art fine-tuning techniques. Furthermore, we use this quality metric along with our diverse prompt selection strategy to construct a new paired preference dataset from scratch. We show that doing preference fine-tuning on this dataset results in a more robust model compared to tuning it on human-labeled preference datasets. For brevity, we refer to our approach as DPO-QSD or QSD. Figure 2 shows the generated image outputs from SDXL base, SDXL-DPO checkpoint fine-tuned on the full Pick-a-Pic v2 dataset (Kirstain et al., 2023) of approximately 1 million image preference pairs, and SDXL-DPO-QSD fine-tuned on top 50 thousand preference pairs selected via our method.

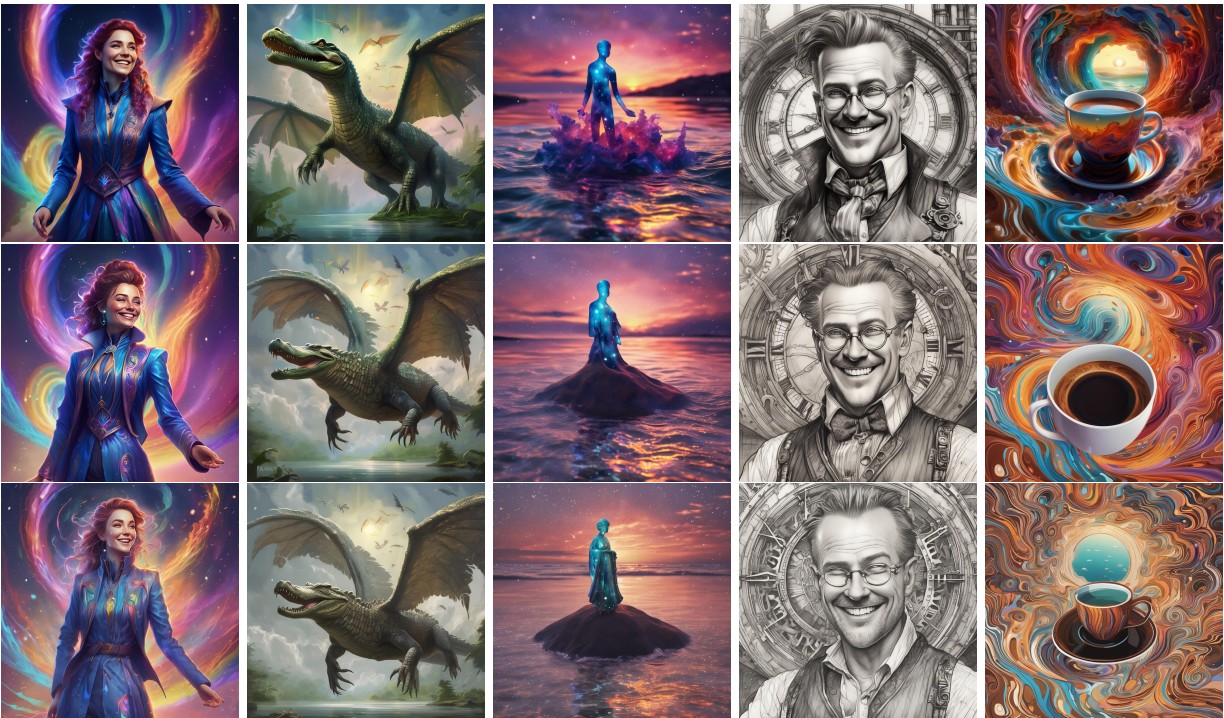

Figure 2: Top to Bottom: *SDXL-DPO-QSD, SDXL-DPO, SDXL*
Prompts: *(1) A smiling beautiful sorceress wearing a high necked blue suit surrounded by swirling rainbow aurora, hyper-realistic, cinematic, post-production (2) Concept art of a mythical sky alligator with wings, nature documentary (3) A galaxy-colored figurine is floating over the sea at sunset, photorealistic (4) close up headshot, steampunk middle-aged man, slick hair big grin in front of gigantic clocktower, pencil sketch (5) A swirling, multicolored portal emerges from the depths of an ocean of coffee, with waves of the rich liquid gently rippling outward. The portal engulfs a coffee cup, which serves as a gateway to a fantastical dimension. The surrounding digital art landscape reflects the colors of the portal, creating an alluring scene of endless possibilities.*

## 2 Related Work

The alignment of diffusion models with human preferences has become a critical area of research, especially as these models are being used increasingly to generate content with downstream objectives. The alignment of diffusion models with human preferences can be categorized into two broad categories: with a reward model and without a reward model. Approaches such as DRAFT(Clark et al., 2024), AlignProp(Prabhudesai et al., 2023), and ReFl(Xu et al., 2023) directly backpropagate the gradients from a differentiable reward to fine-tune the diffusion model. Different reward models are used to fine-tune the diffusion model based on the end-user task. These approaches work for a finite vocabulary set, but do not generalize well to an open vocabulary set and struggle to optimize for complex reward functions such as CLIP score (Radford et al., 2021). DPOK(Fan et al., 2023) and DDPO(Black et al., 2023) are reinforcement learning approaches that maximize the score of the reward model over a set of limited prompts, which limits the performance of these methods as the number of prompts increases. DOODL(Wallace et al., 2023b) attempts to generate more aesthetically pleasing images by making iterative improvements to the generation at run-time; however, this impacts inference efficiency.

The other set of approaches, which do not use an explicit reward model, are inspired by the success of direct preference optimization. The recent work of Diffusion-DPO(Wallace et al., 2023a) is able to fine-tune a diffusion model on a dataset of prompts and image pairs by reformulating the loss function. Diffusion-KTO(Li et al., 2024) builds on top of Diffusion-DPO and does not require pairwise preference data, allowing

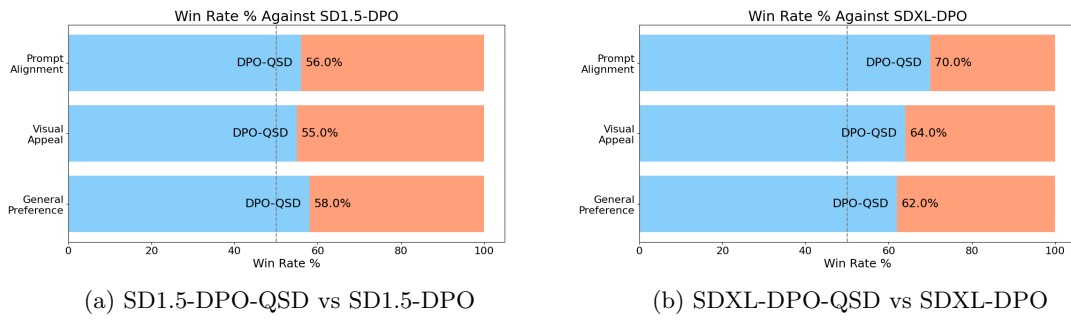

(a) SD1.5-DPO-QSD vs SD1.5-DPO    (b) SDXL-DPO-QSD vs SDXL-DPO

Figure 3: SD1.5 and SDXL QSD models significantly outperform the baseline models in human evaluation.

fine-tuning of diffusion models on single image feedback. However, it still requires a label of winning/losing for each image. Additionally, D3PO(Yang et al., 2023) suggest creating its own image pairs from a set of prompts and then using a reward model to identify preferred images. Despite all these advances, these approaches still suffer from noisy pairwise preference datasets due to human subjectivity.

Most diffusion models (Rombach et al., 2021; Podell et al., 2023; Chen et al., 2023; 2024; Esser et al., 2024) are sometimes trained in two stages, where the first stage involves training on a broad dataset followed by fine-tuning on a carefully selected high quality dataset that is more preferable to humans. This approach involves training the diffusion model on a subset of high-quality images which are selected via an AI reward model, usually an aesthetics classifier. Parrot(Lee et al., 2024) uses Pareto-optimal sorting to rank images on multiple reward scores to select the optimal subset. Models like DALLE-3(Shi et al., 2020), SD3(Esser et al., 2024), and CogView(Ding et al., 2021) re-caption existing web-scraped datasets to improve text fidelity. However, these approaches require large amounts of resources to caption millions of images. This approach of selecting a subset of the data set to better align with human preferences has not been explored in the domain of reinforcement learning for text-to-image models.

## 3 Method

### 3.1 Background

Diffusion-DPO (Wallace et al., 2023a) considers a setting with a dataset $D = \{(c, x_w, x_l)\}$ where each sample consists of a prompt $c$, a winning image $x_w$, and a losing image $x_l$. The aim is to train a new model $p_\theta$ on these preference pairs, which is better aligned with human preferences compared to the reference model $p_{ref}$. Diffusion-DPO achieves this by completely removing the reward model and reformulating the loss as a function that encourages more denoising at $x_w$ than $x_l$.

However, as we can observe from Figure 1a, human preference is subjective, and this sometimes results in noisy labels. Existing approaches do not try to identify these noisy labels and use the entire dataset for fine-tuning as is. For instance, Diffusion-DPO selects all image preference pairs, excluding only those with ties, without any validation of the preferences. To address this, we sought to identify preference pairs that align more closely with overall trends in preference modeling.

### 3.2 Quality Metric for Ranking Preference Pairs

We propose a new quality metric that works on a pair of images, where a higher score indicates a greater likelihood that the pair will be correctly labeled. Consider any paired preference dataset $D = \{(c^1, x_w^1, x_l^1), (c^2, x_w^2, x_l^2), ...., (c^n, x_w^n, x_l^n)\}$, where each sample consists of a caption $(c)$, a winning image $(x_w)$, and a losing image$(x_l)$. We use an AI reward model, trained to capture human preferences, to estimate the probability that the winning image is correctly ranked as the winner and the losing image as the loser. We use the HPS-v2 model that is trained for human preference to output the score for an image given the prompt. These preference values, after normalization, will range from 0-1, allowing us to interpret them

as the probability that an image is preferred. We refer to this model as $\psi$. Now, the quality $Q$ of each sample pair can be written as

$$Q(c, x_w, x_l) = \psi(x_w/c) * (1 - \psi(x_l/c)) \tag{1}$$

This can be viewed as the probability of the pair being correct, i.e. the probability of the winning image being the winning image and the losing image being the losing image.

Through experiments, we demonstrate that fine-tuning with higher-quality pairs leads to improved model performance. However, as lower-quality pairs are introduced, performance begins to decline, supporting the importance of ranking image preference pairs. In Figure 1b, we plot the quality of the paired image data in the Pick-a-Pic v2 train dataset. We see a sharp decrease in quality score for the initial 100 thousand pairs, followed by a gradual decline for the majority of the dataset, and finally another sharp drop towards the end, where the samples are of the poorest quality. This plot illustrates that the dataset has high quality pairs (Figure 7) where the winning image is clearly better, ambiguous pairs (Figure 8) pairs where the preference is more subjective and low quality pairs (Figure 9) where the reward model does not agree with human labels.

### 3.3 Leveraging Quality Metric for Image Generation

Using our quality metric as described above, we set out to create a new high-quality preference dataset that is not subject to human subjectivity.

**Prompt selection strategy**: Let $P$ denote the initial set of prompts, where $P_0$ is a small randomly selected subset of prompts from DiffusionDB dataset (Wang et al., 2023) and the Pick-a-Pic v2 training set. Both of these datasets consist of prompts designed by real users, which ensures that the prompts reflect a wide range of real-world scenarios. Our goal is to iteratively expand $P_0$ while ensuring semantic diversity. To achieve this, we use a pre-trained sentence embedding model $f : P \to \mathbb{R}^{768}$ that maps each prompt $p$ to a 768-dimensional vector space.

For each candidate prompt $p'$, we compute its cosine similarity with all prompts in the current set $P_t$:

$$S(p', P_t) = \max_{p \in P_t} \frac{f(p') \cdot f(p)}{\|f(p')\|\|f(p)\|} \tag{2}$$

If $S(p', P_t)$ is below a predefined threshold $\tau$, the prompt is added to the set:

$$P_{t+1} = P_t \cup \{p'\}, \quad \text{if } S(p', P_t) < \tau \tag{3}$$

We use a $\tau$ of 0.6. This approach ensures that each newly added prompt contains rich semantic information and maintains high diversity. To ensure that this dataset does not contain any NSFW prompts and images, we filter out prompts with an NSFW score greater than 0.5. This process gives us a total of 40022 prompts.

**Image generation strategy**: For preference pair generation, we use multiple diffusion models – SD1.5, SD2.0 and SDXL to generate images. Each model produces 8 images per prompt. We vary the resolution scales across models and use 512x512 for SD1.5, 768x768 for SD2.1 and 1024x1024 for SDXL. We also randomize the generation seed and vary the classifier guidance scale in a range of 3-11. This diversity ensures a comprehensive representation of the text-to-image models' generalization capability. Images generated from the same prompt are paired for comparison. Finally, we randomly select 1 million pairs for preference fine-tuning of the baseline. For our QSD dataset, we sort all these pairs using the quality metric $Q$ and select the top 100 thousand pairs for fine-tuning.

**Dataset Statistics**: The entire dataset consists of 1M samples generated using 40022 prompts with three diffusion models -SD1.5, SD2.1 and SDXL. The distribution of the number of winning/losing images generated by each of the models is shown in Figure 4 left and right respectively. We filter out almost 30% of the prompts with our NSFW filtering. To measure the diversity of the prompts we compute the pairwise cosine

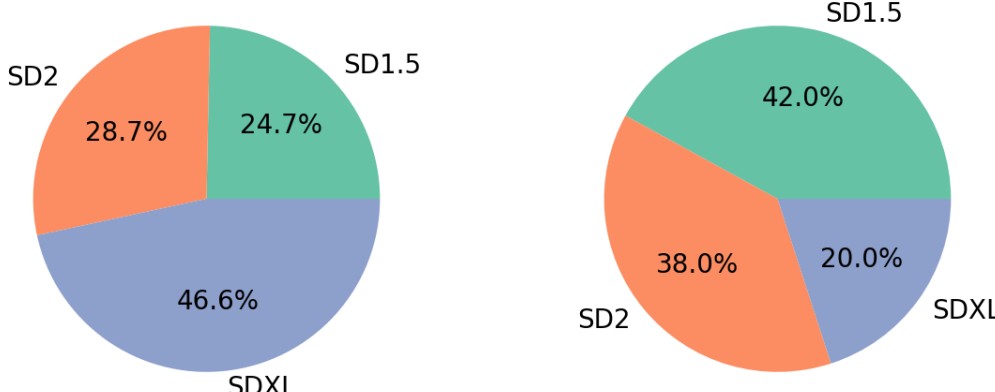

Figure 4: Left: Distribution of models for the Winning image in our synthetic dataset. Right: Distribution of models for the losing image in our Synthetic dataset

similarity between the embeddings of all prompts in the original dataset and compare it to the diverse subset. The average similarity across all the prompts in the original dataset is 0.390, while the average similarity across all the prompts of the diverse subset is 0.272, which indicates that, on average, the prompts in the diverse subset are more distinct from one another.

## 4 Experiments

### 4.1 Dataset

We demonstrate the efficacy of our model on the Pick-a-Pic v2 dataset (Kirstain et al., 2023), which is a crowd-sourced dataset. A human reviewer is presented with a caption and a pair of images generated by T2I models such as Stable Diffusion 2.0 (Rombach et al., 2021), Dreamlike Photoreal 2.05, and Stable Diffusion XL (Podell et al., 2023). The reviewer selects one of the two presented images as more preferred or marks it as a tie. The dataset contains 1 million rows split into 959.5k rows, 20.5k rows, and 20.5k rows of train, validation, and test set, respectively. The training set contains approximately 58k distinct captions.

### 4.2 Model training and Hyper-parameters

**Model training**: We rank the samples in the preference dataset using the quality metric and then use incremental subsets of the dataset for training. To achieve this, we sort the training data by these quality scores and evaluate model checkpoints every 12.5k samples. This process enables us to train on progressively larger subsets of the data within a single training run. We follow this approach to test the impact of adding pairs with decreasing quality as training progresses. For baseline training, we randomly shuffle the dataset.

**Hyperparameters**: We run experiments on SD1.5(Rombach et al., 2021) and SDXL(Podell et al., 2023) models. For pairwise preference fine-tuning we use the fine-tuning approach as highlighted in Diffusion-DPO(Wallace et al., 2023a). For all experiments, we use the ADAMW optimizer with a batch size of 128. All experiments are run on a node of 8 NVIDIA 80 GB A100 GPUs. We train at fixed-square resolution of 512x512 for SD1.5 and 1024x1024 for SDXL. We train for 1 epoch with a learning rate of $1e^{-4}$ for SD1.5 and $1e^{-5}$ for SDXL. In line with the Diffusion-DPO paper, we use a $\beta$ value of 2000 for SD1.5 and 5000 for SDXL. We do not use any dataset augmentations and keep learning rate constant with no warm-up. For all

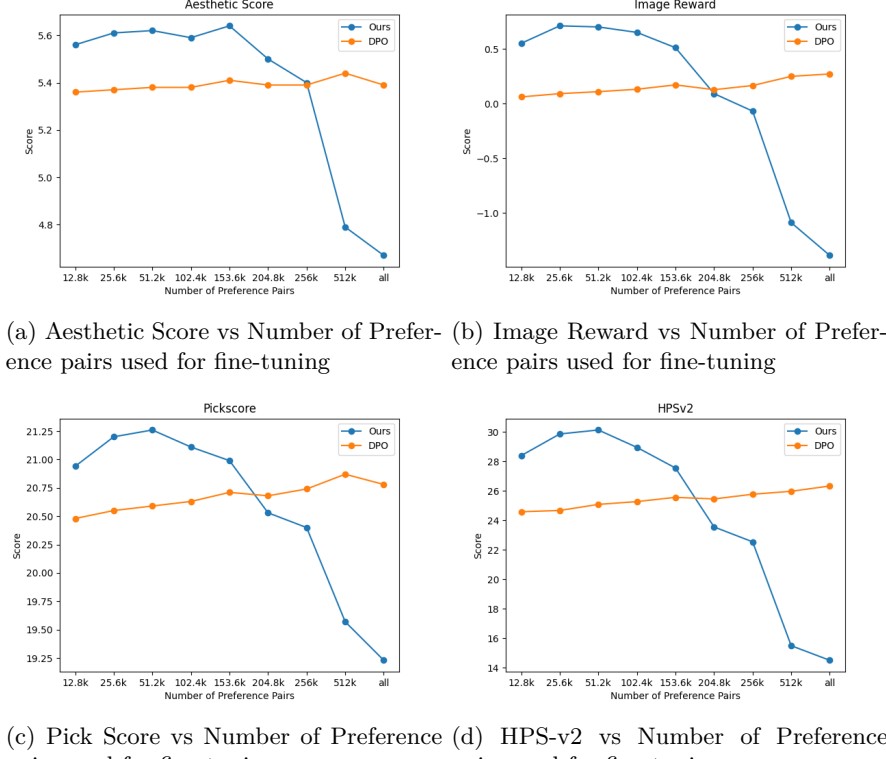

(a) Aesthetic Score vs Number of Preference pairs used for fine-tuning

(b) Image Reward vs Number of Preference pairs used for fine-tuning

(c) Pick Score vs Number of Preference pairs used for fine-tuning

(d) HPS-v2 vs Number of Preference pairs used for fine-tuning

Figure 5: Trend of aesthetic score, Image Reward, PickScore and HPS-v2 while Diffusion-DPO fine-tuning of SD 1.5 on our quality-sorted dataset vs full dataset.

our experiments, we fine-tune using the LoRA approach and use a rank of 64 for both SD1.5 and SDXL. The training seed for all runs is kept constant.

### 4.3 Evaluation

To verify the effectiveness of our approach, we compare with state-of-the-art human preference learning approaches like Diffusion-DPO (Wallace et al., 2023a) fine-tuned on the entire training dataset. As we use the LoRA technique, we also fine-tune LoRAs for the state-of-the-art approaches and compare against them. We evaluate all checkpoints on the Pick-a-Pic validation set (Kirstain et al., 2023), which consists of 500 unique prompts. We choose four AI reward models: ImageReward (Xu et al., 2023), Pickscore, HPS-v2 (Wu et al., 2023), and Laion aesthetics classifier. ImageReward is the first general-purpose text-to-image human preference reward model, which is trained on a total of 137k pairs of expert comparisons. PickScore is a CLIP-based scoring model with a variant of InstructGPT's (Ouyang et al., 2022) reward model objective. Laion aesthetics classifier is also a CLIP based model with a pre-trained MLP that is used to measure the aesthetic quality of an image. We also present scores from the HPS-v2 scoring model on the HPS-v2 test set, which consists of 3200 prompts. HPSv2 is a preference prediction model trained on the HPD-v2 dataset. HPS-v2 can be used to compare images generated with the same prompt. Additionally, we perform a user study to compare our approach to the state-of-the-art Diffusion-DPO. Similarly to Diffusion-DPO, we employ reviewers to select the preferred generation under three different criteria: Q1 General Preference (Which image do you prefer given the prompt?), Q2 Visual Appeal (prompt not considered) (Which image is more visually appealing?) Q3 Prompt Alignment (Which image better fits the text description?). Five responses are collected for each comparison, and the majority vote (3+) is considered the collective decision. For the user study, we randomly sample 25 prompts from each of the four sub-sections of the HPS-v2 test set: photos, anime, paintings, and concept-art.

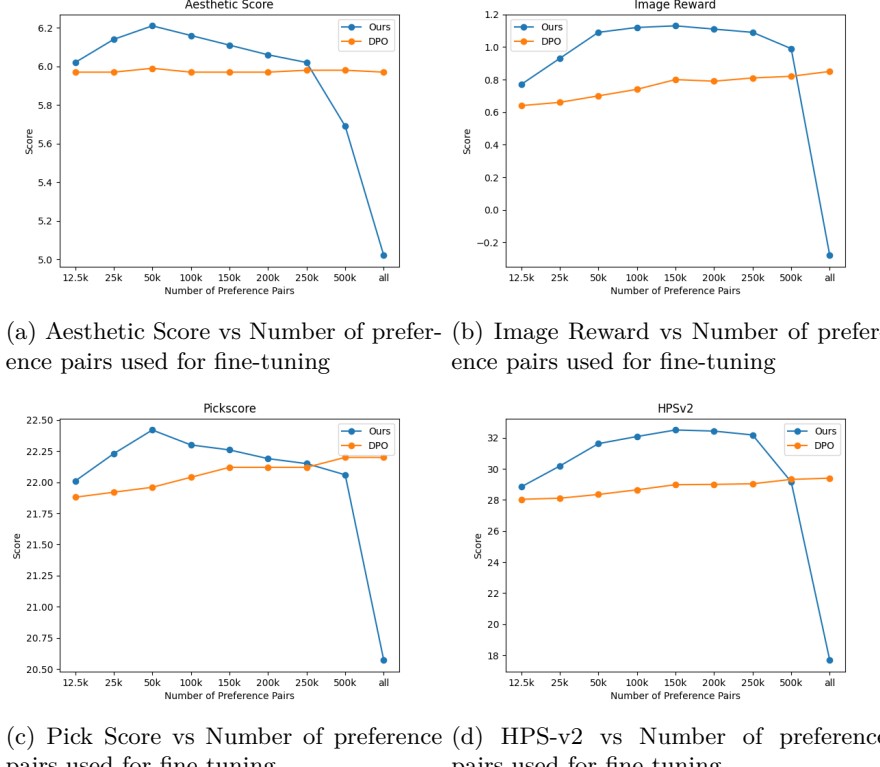

(a) Aesthetic Score vs Number of preference pairs used for fine-tuning

(b) Image Reward vs Number of preference pairs used for fine-tuning

(c) Pick Score vs Number of preference pairs used for fine-tuning

(d) HPS-v2 vs Number of preference pairs used for fine-tuning

Figure 6: Trend of aesthetic score, Image Reward, PickScore and HPS-v2 while Diffusion-DPO fine-tuning of SDXL on our quality-sorted dataset vs full dataset.

## 5 Results

## 6 Results

In Figure 5, we plot the results of training models on incremental subsets as we add more data with decreasing quality. The figure shows that the models fine-tuned using Diffusion-DPO(Wallace et al., 2023a) on the Pick-a-Pic v2 dataset sorted via our quality metric significantly outperform the baseline models fine-tuned using Diffusion-DPO on randomly sampled Pick-a-Pic v2, across four key metrics. These results from the best checkpoints for these training runs are also presented in Table 1. We also observe significant improvements in fine-tuning efficiency with our SD1.5 DPO-QSD model and our SDXL DPO-QSD model outperforming the baseline models with only 5.33% of the data sorted using our metric. As our fine-tuning data increases, we see a peak in the performance of both models after which the metrics start decreasing or start plateauing. This proves our initial hypothesis that not all fine-tuning pairs are equal and that some fine-tuning data does more harm than good by sending adverse signals. Using only 5.33% of the Pick-a-Pic v2 dataset we achieve our best models, which vastly outperform the baseline models fine-tuned on the full training dataset. This also shows that over 90% of the preference pairs in the Pick-a-Pic v2 training set negatively impact training and can be discarded. For our synthetic dataset we observe similar trends with significant gains in all metrics while using only 10% of the dataset selected via our quality metric. Another important observation is that the baseline and final models trained using our synthetic data outperform the models trained using the Pick-a-Pic v2 dataset which further cements the practical application of our synthetic data generation method.

Similarly, the user study in Figure 3 shows that our models are preferred by human raters over the baseline Diffusion-DPO models. Our SDXL DPO-QSD model is preferred by human annotators 70% of the time in

Table 1: Comparison of our DPO-QSD approach with baseline DPO for SD1.5 and SDXL on the Pick-a-Pic v2 dataset and for SD1.5 on the synthetic dataset.

| Method | Dataset | Aesthetic | ImageReward | PickScore | HPSv2 | % Training Data |
|---|---|---|---|---|---|---|
| SD1.5 DPO | Pick-a-Pic | 5.39 | 0.27 | 20.78 | 26.34 | 100% |
| **SD1.5 DPO-QSD** | Pick-a-Pic | **5.62** | **0.70** | **21.26** | **30.14** | **5.33%** |
| SD1.5 DPO | Synthetic | 5.45 | 0.31 | 20.75 | 25.99 | 100% |
| **SD1.5 DPO-QSD** | Synthetic | **5.77** | **0.64** | **21.02** | **28.54** | **10%** |
| SDXL DPO | Pick-a-Pic | 5.97 | 0.85 | 22.20 | 29.40 | 100% |
| **SDXL DPO-QSD** | Pick-a-Pic | **6.21** | **1.09** | **22.42** | **31.62** | **5.33%** |

Table 2: Efficacy of our ranking method on different fine-tuning paradigms using Pick-a-Pic dataset. The results prove that our ranking approach improves performance and training efficiency across fine-tuning approaches.

| Method | Aesthetic | ImageReward | PickScore | HPSv2 | % Training Data |
|---|---|---|---|---|---|
| SLIC-HF baseline | 5.45 | 0.33 | 20.93 | 26.71 | 100% |
| **SLIC-HF-QSD** | **5.69** | **0.72** | **21.24** | **29.65** | **5.33%** |
| ORPO baseline | 5.51 | 0.30 | 20.57 | 26.97 | 100% |
| **ORPO-QSD** | **5.60** | **0.60** | **20.80** | **28.25** | **10.6%** |

prompt alignment, 64% of the time in visual appeal, and 62% of the time in general preference. Similarly, our SD1.5 DPO-QSD model is preferred by human annotators 54% of the time in prompt alignment, 55% of the time in visual appeal and 58% of the time in general preference. This shows that the model generalizes better to human preference and does not capture the biases of the inherent reward model used to scoring the pairs.

# 7 Ablation Studies

## 7.1 Efficacy with different fine-tuning methods

We fine-tune the base model using different fine-tuning methods to show that our QSD is effective in improving performance across different fine-tuning approaches. For all experiments, we fine-tune the baseline on the Pick-a-Pic v2 training dataset with random sampling, while our approach uses the Pick-a-Pic v2 quality-sorted dataset. We experiment with the Diffusion ORPO loss and the SLIC-HF loss (Zhao et al., 2023). We perform this ablation using the LoRA approach for SD1.5 with rank 64, a batch size of 128, and a learning rate of 1e-4. For Diffusion-ORPO inspired by ORPO(Hong et al., 2024), we use a learning rate of 1e-3 for both models.

For these experiments, we select the best performing model and present the results in table 2. As we can observe, our approach performs considerably better than the baseline in both methods. Moreover, our approach achieves these results while using only the top 5.33% of the data for SLIC-HF and the top 10.6% of the data for ORPO, demonstrating a 10x gain in fine-tuning efficiency. This ablation proves that our pair-ranking method improves performance across different fine-tuning paradigms and is not limited to the Diffusion-DPO loss formulation. We believe that the loss in efficiency for Diffusion-ORPO stems from the inclusion of the mean squared error loss of the winning image in the overall loss function, which dominates the other loss terms.

Table 3: Generalization ability of different scoring models. As we can observe, model trained on just 5.33% of pairs ranked best using HPS-v2 greatly outperform the baseline trained on 100% of the data.

| Method | Aesthetic Score | Image Reward | PickScore | HPSv2 | % Training Data |
|--------|-----------------|--------------|-----------|-------|-----------------|
| Baseline DPO | 5.39 | 0.27 | 20.78 | 26.34 | 100% |
| Image Reward | 5.40 | 0.32 | 20.88 | 26.91 | 100% |
| Laion Aesthetics | **5.80** | 0.49 | 21.09 | 27.30 | 16% |
| PickScore | 5.44 | 0.38 | 21.05 | 27.52 | **5.33%** |
| HPS-v2 | 5.62 | **0.70** | **21.26** | **30.14** | **5.33%** |

Table 4: Effect of LoRA rank on training efficiency and model performance. Despite different LoRA sizes, we get our best model at 5.33% of the data, showing that the best selected pairs are independent of model size.

| Method | Rank | Aesthetic | Reward | PickScore | HPSv2 | % Training Data |
|--------|------|-----------|--------|-----------|-------|-----------------|
| Baseline DPO | 32 | 5.43 | 0.25 | 20.93 | 26.39 | 100% |
| DPO-QSD | 32 | **5.58** | **0.68** | **21.24** | **29.82** | **5.33%** |
| Baseline DPO | 64 | 5.39 | 0.27 | 20.78 | 26.34 | 100% |
| DPO-QSD | 64 | **5.62** | **0.70** | **21.26** | **30.14** | **5.33%** |
| Baseline DPO | 256 | 5.42 | 0.35 | 20.91 | 26.65 | 100% |
| DPO-QSD | 256 | **5.66** | **0.70** | **21.24** | **30.06** | **5.33%** |

## 7.2 Preference generalization of different scoring models

We test the preference biases of various scoring models by using different reward models to score each pair of images. For this experiment, we try four different scoring models $\psi_z(c, x_w)$ - HPS-v2, Laion aesthetic score , PickScore, and ImageReward. Since these models output scores on different scales, we normalize them. We standardized the PickScore and clipped the values to +/- 3 which removes the outliers beyond 99% values then shift them to 0-1 by adding 3 and divide with 6. Aesthetic score is divided by 10. Image reward is normalized similarly to PickScore. We run this ablation on the Pick-a-Pic v2 dataset using LoRA for SD1.5 with rank 64, learning rate 1e-4, and a batch size of 128.

We present the results in Table 3. As observed, fine-tuning the model on pairs ranked using HPS-v2 consistently improves performance across all metrics, highlighting its generalization ability. In contrast, the Laion aesthetic predictor, while excelling in aesthetic score, is inherently biased toward aesthetics and fails to generalize across other metrics. PickScore, trained on the Pick-a-Pic v2 dataset, inherits biases from its human-generated labels, making it sensitive to the noise in these annotations. ImageReward, on the other hand, struggles as an effective ranking metric.

## 7.3 Effect of LoRA rank

To test the effect of capacity of the LoRA layers and their effect on the model's ability to learn the new information from the dataset, we run experiments with different dimensions of the LoRA layers. Specifically, we want to see how the performance of the model and the fine-tuning efficiency varies with our QSD dataset as we vary the LoRA rank. We run this experiment on the Pick-a-Pic v2 dataset using lora for SD1.5 as the base model with a learning rate of 1e-4 and a batch size of 128. To this end, we fine-tune with three different LoRA ranks - 32, 64 and 256. For comparison with the baseline, we fine-tuned DPO models with the same hyperparameters and ranks. We present the results in Table 4. As we can observe, we achieve the best results with rank 256 LoRA; however, the improvements over rank 64 are minimal. Therefore, we decide to use rank 64 for our main results. The key observation is that, despite the capacity of the LoRA, we get the best fine-tuning efficiency with only 5.33% of the data.

## 8 Limitations and Future Work

In this work, our focus was primarily on training a better preference-optimized model by using a higher-quality subset of the dataset and using this approach to synthesize a better paired preference data for future use. However, we acknowledge that there are limitations to our approach. Our approach relies on a single quality metric to select training pairs. This makes our approach prone to the biases of the underlying model used to calculate quality. While we discard the ambiguous preference pairs, future works could find a way to leverage this data as well without any performance loss. Future works could also explore the use of the quality metric as a dynamic threshold.

## 9 Conclusion

In this paper, we address the problem of optimal fine-tuning of diffusion models to better align them with human preferences. Unlike previous approaches, we solve this problem by introducing a quality metric that prioritizes high-quality preference pairs and helps to remove low-quality pairs from datasets labeled by human annotators. We also introduce an automated pipeline that leverages our diverse prompt selection strategy and our quality metric to synthesize a high-quality preference dataset free of human subjectivity. We demonstrate that our data selection strategy significantly enhances diffusion model alignment, achieving superior results across multiple AI-based metrics and human evaluators. Our experiments show that models fine-tuned with less than top 10% of the Pick-a-Pick v2 dataset outperform baseline models in both quantitative metrics and human preference evaluations, which highlights the underlying noise in current open-source preference datasets. We run multiple ablations to showcase the efficiency of our data ranking approach across multiple methods. We also demonstrate the effectiveness of our automatic data creation pipeline. The resulting models trained on this dataset surpass the models trained on open-source datasets. Thus, we validate our initial hypothesis that not all preference pairs contribute equally, and fine-tuning on the entire dataset can be detrimental. By applying our fine-tuning strategy alongside early stopping, one can significantly enhance training efficiency, leading to a more robust and powerful model.

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

# A  Appendix

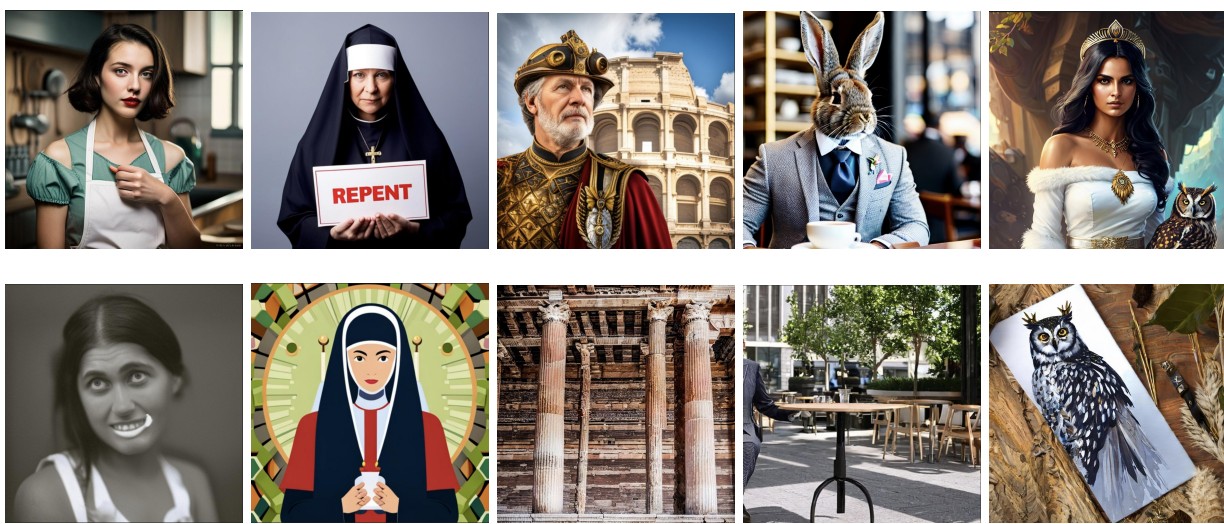

Figure 7: Examples of good pairs ranked best using our method. *Top row: Winning Image, Bottom row: losing image.* The winning images of good samples have better prompt adherence, aesthetic score and are more preferable to humans. Caption from left to right: *(1) A closeup portrait of a playful maid, undercut hair, apron, amazing body, pronounced feminine features, kitchen, freckles, flirting with camera, (2) A nun holding a sign that says repent, (3) Roman emperor, photo, palace background, (4) A rabbit in a 3 piece suit, sitting in a cafe. Hyper Realistic, ultra realistic, 8k, (5)a painting of a woman with an owl on her shoulder, james gurney and andreas rocha, owl princess with crown, also known as artemis or selene, wlop and sakimichan, detaild, portrait character design, falcon, portrait of modern darna, crowned, golden goddess, white witch, by Johannes Helgeson, goddess of travel*

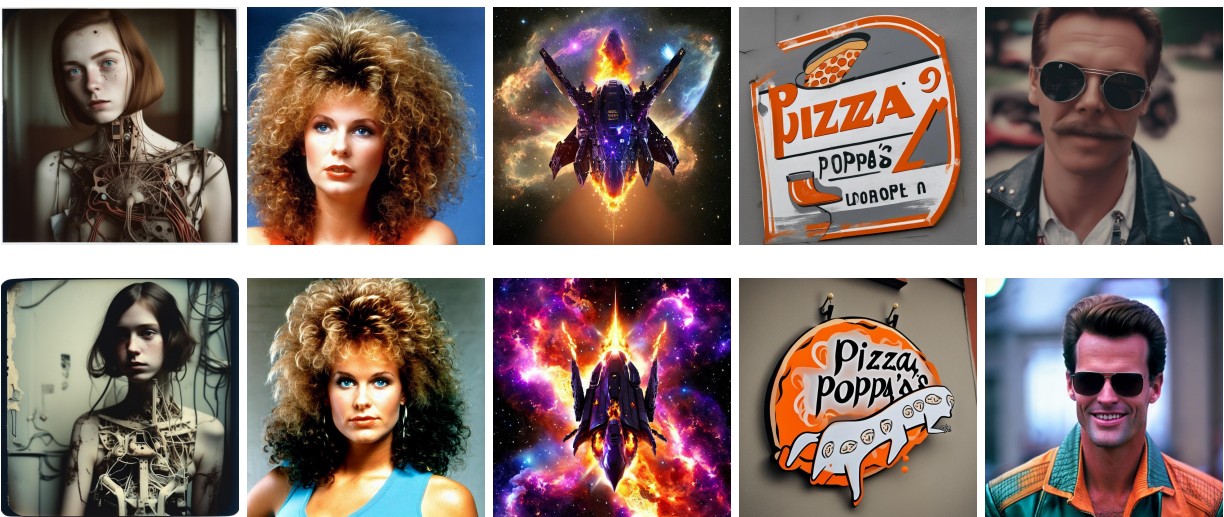

Figure 8: Examples of ambiguous pairs identified using our method. Top row: *Winning Image* | Bottom row: *Losing image* | Captions: **a)** *polaroid photo, abandoned room, pale thin young woman with freckles made of complex machine with wires and intestines, dust, cobwebs* **b)** *young woman with big teased 80's hair* **c)** *void space ship galaxy on fire* **d)** *The image features a close-up of a sign for a pizza shop. The sign is orange and white, with the words "Pizza" and "Poppa's" visible. The sign is hanging on a grey wall. The close-up view emphasizes the colors and details of the sign, making it visually appealing and attention-grabbing for potential customers passing by.* **e)** *A sharp close up selfie of a man in the 1980s era. He's happy, stylish. Wears a jacket for motorcycle drivers. You can feel the 80's feeling. Made with an old camera style, there's that 80's old camera color filter. Sharp, clear quality. 4k. Synthwave, retrowave feel. You can see his hand reach out of the photo - like he would hold his camera out of the frame. His face is tilted towards his shoulder*

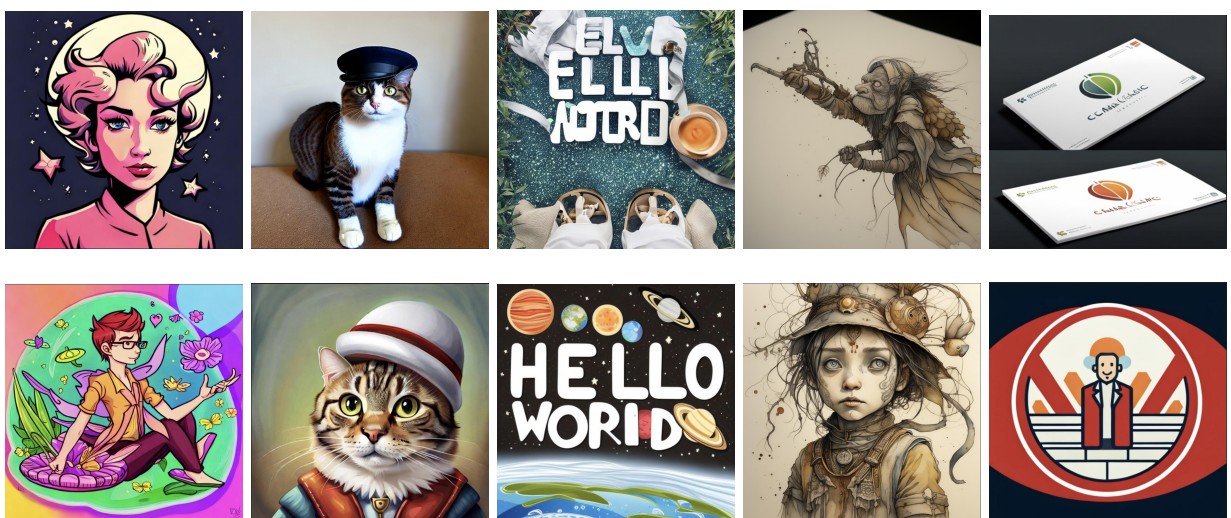

Figure 9: Examples of bad pairs identified by our method. *Top row: Winning Image, Bottom row: losing image.* As can be observed, in these pairs the losing image is better in some quality like aesthetics or prompt adherence over the winning image. Caption from left to right: *(1)a little faery floating in the style of dan hipp, (2) cat wearing a hat, (3) "Hello world" text, space, planets style, (4) face close up woman Jean-Baptiste Monge, watercolour and ink, intricate details, a masterpiece, dynamic backlight, (5) Design a logo for a modern, high-end medical clinic that specializes in personalized, holistic healthcare. The clinic is called "C" and focuses on improving patients' overall well-being through nutrition, exercise, and mental health support. The logo should be simple, sleek, and convey a sense of warmth and approachability while still exuding professionalism and expertise*

