# OpenReview forum: "Effective Text-to-Image alignment with Quality Aware Pair Ranking"
_TMLR — Rejected by TMLR_

### Review · Reviewer_ognY · 2025-10-16

**Summary Of Contributions:**

The main contributions are as follows:
(i) The authors propose a quality metric to rank image preference pairs and achieve more effective Diffusion-DPO fine-tuning.
(ii) The authors use the quality metric and a prompt selection strategy to generated a new synthesized dataset for training.

Strength:
1. The proposed quality metric could help reduce the training resources while achieving better performance.
2. The authors provide comprehensive experimental results on various sub-set proportion with different scoring functions.

Weakness:
1. The technical contribution is limited. The main contributions are (i) propose a ranking metric and (ii) use this metric to generate a synthetic dataset.
2. The motivation is not clearly presented. In the abstract the authors start from the problem that preference datasets are usually labeled by human annotators, which are inherently subjective and prone to noisy labels. However, this should motivate the second contribution for generating the synthetic dataset. Developing a quality metric to rank image preference pairs does not directly address this problem.
3. The authors do not present enough analysis on the proposed quality metric (Equation 1). Why is the metric in this form? Are there any other possible metric? A simple baseline to compare is just using existing reward scores to select top 5% or 10% data for fine-tuning. Why is Equation 1. the most appropriate metric?
4. The statement that "fine-tuning on this new dataset achieves better results than the models trained using human labeled datasets" (in abstract) is not solid. The authors only discuss this contribution with one sentence in the results section. From table 1 it seems that the conclusion is not well-supported. For example, It seems that PickScore and HPSv2 both decrease for SD1.5 DPO with 100% training data. Additionally, it is not enough to show two sets of experiments and conclude the proposed synthetic dataset is better. Additional experiments with more baselines are needed.
5. It is not clear the criterion to select the number of subset samples for training. For example in Table 2, it is not clear why it is 5.33% for SLC-HF-QSD and why 10.6% for ORPO-QSD.

**Audience:**

Yes

**Audience Explanation:**

This paper would be interesting to the researchers working on Direct Preference Optimization. It provides insights on finding optimal data for more effective training.

**Claims And Evidence:**

No

**Claims Explanation:**

See Summary Of Contributions - Weakness 4. The statement that "fine-tuning on this new dataset achieves better results than the models trained using human labeled datasets" (in abstract) is not solid.

**Requested Changes:**

1. As is mentioned in weakness 2, the authors should consider revise the abstract and introduction to better motivate the problems to solve and corresponding contributions.
2. As is mentioned in weakness 3, the authors should consider add additional comparison with different possible quality metrics.
3. As is mentioned in weakness 4, to claim that the generated synthesis dataset is better, the authors should add additional baselines and compare the performance between trained on Pick-a-Pic dataset and proposed dataset.
4. As is mentioned in weakness 5, the authors should consider  clarify how the size of the sub data sets are determined.
5. When referring to Figure 1, the authors claim "with a large portion of the samples exhibiting higher scores for the 'losing' images compared to their assigned 'winning' counterparts". It is not clear in Figure 1 where this comparison between losing images and winning images are illustrated.
6. It is not clear in Figure 5 and Figure 6, why the performance of DPO models (orange line) always increase, if the authors claim that
"over 90% of the preference pairs in the Pick-a-Pic v2 training set negatively impact raining and can be discarded".

Other minor revision points
- Section 5 Results is empty and duplicated with Section 6.
- Figure 3 is first mentioned on page 8 but appears on page 4. The authors should consider re-arrange the images.

---

### Review · Reviewer_eXkT · 2025-10-17

**Summary Of Contributions:**

This paper argues the existing approaches that align the text-to-image model with human preferences are prone to human annotator’s subjectivity and noisy labels. The paper hypothesis the model trained on such data will be negatively affected. To address this issue, this paper proposed a metric that evaluates and ranks the preference pairs in a dataset, to identify clear and strong preferences as opposed to ambiguous or subjective choices. With such score, the paper also constructed a dataset that supposed to prioritize better quality pairs. Quantitative and ablation study are reported to support the effectiveness claim of the proposed metric.

### Strength

1. The main observation about that the subjectiveness and noisy label could have negative impact on model training is sensible.
2. The reported quantitative experiments and ablation study shows the potential of the proposed approach.
3. Limitations of the proposed approach are also discussed in the paper.

### Weakness

1. Lack of Direct Evidence for the Core Hypothesis. The paper's central hypothesis is that low-quality preference pairs (those with similar scores or labels that conflict with reward models) are detrimental to training. However, the paper does not provide a direct, standalone analysis to validate this hypothesis *before* presenting its main contribution. Instead, the proof is offered indirectly, by showing that a model trained *without* these pairs performs better,  which is a "proof by successful application" rather than a "proof by extensive analysis.”
2. Propagated Noise from the HPS-v2 Reward Model. The proposed quality metric Q is critically dependent on the HPS-v2 reward model. Since HPS-v2 was itself trained on human preference data, it has inevitably learned some of the same biases and noise inherent in that process. The paper does not sufficiently address why using a potentially noisy reward model to filter a noisy dataset is a robust solution, and how the model's own errors don't simply replace the original dataset's noise.
3. The Role of Hard Examples in Learning. The methodology explicitly trains on 'easy' examples where the preference is clear, while discarding ambiguous or 'hard' pairs where image qualities are similar. Conventional machine learning wisdom suggests that training on such hard examples near the decision boundary is crucial for building a robust model. The paper doesn't justify why discarding them is beneficial.
4. Multiple method / evaluation details are missing.
    - In Section 3.3, the prompt selection strategy relies on a pre-trained sentence embedding model to ensure diversity. However, the specific model used is not identified, which harms the reproducibility of the synthetic dataset creation process.
    - The methodology involves filtering prompts using an "NSFW score," but the tool, model, or service used to generate this score is not specified, again making this step non-reproducible.
    - The paper states LoRA is used for fine-tuning but may lack sufficient detail on its specific implementation, such as which network modules (e.g., attention layers) were adapted.
    - Section 4.3 describes a user study but omits crucial details needed to assess its validity, such as the number of unique human reviewers, their level of expertise (e.g., crowd-workers vs. authors), and the interface used.

**Audience:**

Yes

**Audience Explanation:**

The topic "text to image model alignment" is well within the scope of TMLR imo.

**Claims And Evidence:**

No

**Claims Explanation:**

See Weaknesses: 1, 2, 3

**Requested Changes:**

Please address the Weaknesses.

---

### Review · Reviewer_XgWM · 2025-10-23

**Summary Of Contributions:**

### **Summary**

This paper introduces a novel approach for improving the alignment of text-to-image (T2I) diffusion models with human preferences by addressing the issue of noisy or subjective labels in human-annotated preference datasets. The core contribution is the **Quality-Sorted Dataset (QSD)** — a ranking-based fine-tuning framework that leverages a **quality metric** derived from neural reward models (notably HPS-v2) to select only the most reliable preference pairs for effective RL-based fine-tuning method requires preference pair (Diffusion-DPO).

Key contributions include:

**1. Quality Metric for Pair Ranking**: A probabilistic metric $\( Q(c, x_w, x_l) = \psi(x_w \mid c) \times (1 - \psi(x_l \mid c)) \)$
 estimating how confidently one image in a pair is better than the other.


**2. Quality-Sorted Dataset (QSD)**: A dataset ranking strategy that fine-tunes diffusion models on top-quality preference pairs (as little as 5.33% of total data) and achieves higher quantitative and qualitative performance than training on full datasets.


**3. Synthetic Dataset Generation for effective alignment**: A data synthesis pipeline combining diverse prompt selection and multi-model generation (SD1.5, SD2.1, SDXL) to create 1M new preference pairs ranked by the proposed metric, outperforming human-labeled datasets.


**4. Empirical Validation**: Demonstrated that fine-tuning with top-ranked subsets are efficient and leads to significant improvements across Aesthetic Score, PickScore, ImageReward, and HPS-v2 metrics, as well as human evaluation benchmarks.

**Audience:**

Yes

**Audience Explanation:**

The topic of improving text-to-image alignment through data quality ranking is timely and relevant to researchers in generative modeling, diffusion or language model alignment.

**Claims And Evidence:**

Yes

**Claims Explanation:**

The experimental results are comprehensive and well-documented, showing consistent quantitative and qualitative improvements across multiple metrics and models.

**Requested Changes:**

**1. Clarify dataset separation and result presentation:**
- The current Evaluation and Results section mixes outcomes from Pick-a-Pic v2 and the synthetic dataset. Please separate these experiments more explicitly; e.g., divide Table 1 into two parts (Pick-a-Pic vs. Synthetic) and clarify in the text which dataset each figure corresponds to.
- In particular, include **SDXL results on the synthetic dataset**, since Figure 6 currently only shows SDXL performance on Pick-a-Pic v2. This addition is crucial for evaluating the scalability and general applicability of the proposed Quality-Sorted Dataset (QSD) framework.
- Finally, provide an **ablation study on the proportion of synthetic data** used for fine-tuning (e.g., 1%, 5%, 10%, 20%), similar to the incremental subset analysis performed for Pick-a-Pic v2. This would justify the choice of 10% as the optimal threshold and demonstrate the stability of QSD’s performance on synthetic data.


**2. Address evaluation bias and potential overfitting:**
- The paper relies heavily on the HPS-v2 reward model, both for ranking preference pairs and as an evaluation metric. This design risks circularity and may overstate performance improvements.
- Please include additional evaluation using an independent, **non-CLIP-based** metric (e.g., VQA-Score [R1] or VLM-based evaluation [R2]) to demonstrate that the observed gains reflect broader preference alignment rather than overfitting to the ranking metric.


**References**
1. Liu et al., Evaluating Text-to-Visual Generation with Image-to-Text Generation. ECCV 2024. (https://arxiv.org/abs/2404.01291)
2. Tan et al., EvalAlign: Supervised Fine-Tuning Multimodal LLMs with Human-Aligned Data for Evaluating Text-to-Image Models (https://arxiv.org/abs/2406.16562)

---

### Decision · Action_Editor_XVcu · 2025-11-20

**Recommendation:** Reject

**Audience:**

Yes

**Audience Explanation:**

The reviewers view the topic is of interest to the TMLR's audience.

**Claims And Evidence:**

No

**Claims Explanation:**

The reviewers identified several weaknesses in the contribution, methodology, and evaluation. The authors did not provide a rebuttal addressing these concerns.